Species level mapping of a seagrass bed using an unmanned aerial vehicle and deep learning technique

Tahara Satoru 1
Sudo Kenji 2 3
http://orcid.org/0000-0002-5451-1231 Yamakita Takehisa 4
Nakaoka Masahiro 2 nakaoka@fsc.hokudai.ac.jp
1 Graduate School of Environmental Science, Hokkaido University , Sapporo, Hokkaido , Japan
2 Akkeshi Marine Station, Field Science Center for Northern Biosphere, Hokkaido University , Akkeshi, Hokkaido , Japan
3 Japan Fisheries Research and Education Agency, Fisheries Technology Institute , Hatsukaichi, Hiroshima , Japan
4 Marine Biodiversity and Environmental Assessment Research Center (BioEnv), Research Institute for Global Change (RIGC), Japan Agency for Marine Earth Science and Technology , Yokosuka, Kanagawa , Japan
Provete Diogo
Electronic publication date: 2022 Oct 17
Publication date: 2022
Volume: 10
Electronic Location ID: e14017
Received 2022 Feb 28; Accepted 2022 Aug 16
Copyright: © 2022 Tahara et al.
Copyright year: 2022
Copyright holder: Tahara et al.
License: This is an open access article distributed under the terms of the Creative Commons Attribution License, which permits unrestricted use, distribution, reproduction and adaptation in any medium and for any purpose provided that it is properly attributed. For attribution, the original author(s), title, publication source (PeerJ) and either DOI or URL of the article must be cited.
License URL: https://creativecommons.org/licenses/by/4.0/

Keywords: Remote sensing, Spatial mapping, Species identification, Drone, Deep neural network, Zostera marina, Zostera japonica, Accuracy assessment, Hokkaido, Japan

Funding: Saroma-ko Aquaculture Cooperation, Japan Society for the Promotion of Science 16H01792 Environmental Restoration and Conservation Agency (Environment Research and Technology Development Fund), S-15 Predicting and Assessing Natural Capital and Ecosystem Services (PANCES) SATREPS program by the Japan International Cooperation Agency (JICA) and the Japan 901 Science and Technology Agency (JST) This work is supported by a fund from Saroma-ko Aquaculture Cooperation, Japan Society for the Promotion of Science (Kakenhi 16H01792), Environmental Restoration and Conservation Agency (Environment Research and Technology Development Fund), S-15 Predicting and Assessing Natural Capital and Ecosystem Services (PANCES), and the SATREPS program by the Japan International Cooperation Agency (JICA) and the Japan 901 Science and Technology Agency (JST). The funders had no role in study design, data collection and analysis, decision to publish, or preparation of the manuscript.

==============================
Background

Seagrass beds are essential habitats in coastal ecosystems, providing valuable ecosystem services, but are threatened by various climate change and human activities. Seagrass monitoring by remote sensing have been conducted over past decades using satellite and aerial images, which have low resolution to analyze changes in the composition of different seagrass species in the meadows. Recently, unmanned aerial vehicles (UAVs) have allowed us to obtain much higher resolution images, which is promising in observing fine-scale changes in seagrass species composition. Furthermore, image processing techniques based on deep learning can be applied to the discrimination of seagrass species that were difficult based only on color variation. In this study, we conducted mapping of a multispecific seagrass bed in Saroma-ko Lagoon, Hokkaido, Japan, and compared the accuracy of the three discrimination methods of seagrass bed areas and species composition, i.e., pixel-based classification, object-based classification, and the application of deep neural network.

Methods

We set five benthic classes, two seagrass species (Zostera marina and Z. japonica), brown and green macroalgae, and no vegetation for creating a benthic cover map. High-resolution images by UAV photography enabled us to produce a map at fine scales (<1 cm resolution).

Results

The application of a deep neural network successfully classified the two seagrass species. The accuracy of seagrass bed classification was the highest (82%) when the deep neural network was applied.

Conclusion

Our results highlighted that a combination of UAV mapping and deep learning could help monitor the spatial extent of seagrass beds and classify their species composition at very fine scales.

Introduction

Seagrasses are angiosperms that inhabit relatively shallow environments along tropical and subarctic coasts, and about 60 species are known worldwide (Short et al., 2007). Seagrasses usually form beds composed of single or multiple species. While seagrass beds play an essential role in providing valuable ecosystem services, they are declining in many parts of the world due to natural and human-induced disturbances (Short & Wyllie-Eciieverria, 1996; Waycott et al., 2009; Sudo et al., 2021). Since seagrass distribution and abundance show significant spatiotemporal variability (Tomasko et al., 2005), long-term monitoring of spatial information at each location is essential for appropriate management.

Monitoring of seagrass beds has been conducted using ground-based field surveys (Short et al., 2006), optical remote sensing with aircraft (Kendrick et al., 2000; Sherwood et al., 2017) and satellites (Xu et al., 2021; Zoffoli et al., 2021), and acoustic remote sensing (Gumusay et al., 2019). Field surveys can provide detailed information on seagrass cover, species composition, and biomass. However, they are time-consuming and labor-intensive, and the survey area is limited. In contrast, remote sensing methods can obtain large/wide areal distribution information with less effort than field surveys. In addition, it is possible to analyze long-term temporal changes by using aerial photographs (Yamakita, Watanabe & Nakaoka, 2011). While many results have also been reported using satellite data for long-term monitoring (Lyons, Phinn & Roelfsema, 2012; Calleja et al., 2017; Zoffoli et al., 2020; Xu et al., 2021), several limitations have been pointed out for traditional optical remote sensing. The biggest problem is the resolution. The most commonly used satellite data, the Landsat series, provides data over a wide area at a low cost but has a spatial resolution of 30 m which is too low compared to detailed fine-scale information obtained by in-situ field surveys. Phinn et al. (2008) has reported that higher spatial and spectral resolutions are needed for more accurate detailed mapping. Studies using commercial high-resolution satellite images such as WorldView-2 and RapidEye have reported high mapping accuracy (Coffer et al., 2020). However, these commercial satellite images are too expensive for long-term, broad-scale monitoring.

In recent years, UAVs (unmanned aerial vehicles, or drones) have been increasingly used in field research due to some advantages compared with conventional remote sensing (Nowak, Dziób & Bogawski, 2019). High spatial resolution data are available by low altitude UAV flights. Frequent flight is possible because the no-cloud sky is unnecessary, like satellite, and operation cost is low. It is also possible to adjust the survey time and day, which is impossible with satellites in a fixed orbit. In seagrass research, UAVs have been used for detailed bed mapping (Duffy et al., 2018; Nahirnick et al., 2019; Hobley et al., 2021). Nonetheless, most of these studies mapped seagrass beds consisting of only a single species or conducted mapping without species discrimination.

Seagrasses have different morphologies and life histories depending on the species (Duarte, 1991), and when they live nearby, mapping them by species is necessary to obtain more accurate information such as estimating biomass (Knudby & Nordlund, 2011). Different species provide different ecosystem services (Mtwana Nordlund et al., 2016) and respond differently to changes in the environment (Roca et al., 2016). Thus, developments of detailed methods that can discriminate different seagrass species are promising not only for more effective monitoring of seagrass beds, but also for monitoring and managing invasive species (Kumar et al., 2019).

Few studies performed species discrimination of seagrasses with UAV images. Román et al. (2021) showed that seagrass bed mapping, including seagrass discrimination, can be performed with high accuracy using a UAV-mounted ten bands multispectral camera and automatic classification based on machine learning algorithms. Chayhard et al. (2018) showed that visual interpretation could be applied to classify seagrass species with different morphology, such as long leaves type (Enhalus acoroides) and short leaves type (Halodule pinifolia and H. uninervis), even using the RGB images taken by UAVs. The camera installed in the consumer-grade UAV is an RGB sensor, and the use of a multispectral camera is costly. Therefore, there is a need for developing methods for seagrass species discrimination using image data with limited spectral resolution but high spatial resolution.

In general, spatial distribution mapping of seagrass beds by optical remote sensing is carried out using classification algorithms (Diesing, Mitchell & Stephens, 2016). Classification algorithms classify the image into several classes such as seagrass, bare sand, and macroalgae by computer. They can be divided into supervised classification and unsupervised classification depending on whether training data are used or not. In supervised classification, which uses ground-truth data obtained from field surveys as training data, there are two types of classification: (1) pixel-based classification which classifies each pixel, and (2) object-based classification which classifies each object by grouping similarly colored neighboring pixels. It has been reported that object-based classification provides higher accuracy for high spatial resolution images than pixel-based classification (Gao & Mas, 2008). These classification methods have been used to analyze optical remote sensing data based only on limited image information such as the color, object shape, and size. On the other hand, in ultra-high-resolution UAV images, more features are available, such as the pattern, texture, and location of the objects in the image. A deep neural network (DNN) can automatically extract these various features using a convolutional neural network (CNN), the basic network used for DNN image processing (Traore, Kamsu-Foguem & Tangara, 2018). Seagrass mappings using UAV images and DNN have been conducted in recent years, but they are limited to single species or discriminating seagrass from macroalgae (Hobley et al., 2021; Jeon et al., 2021). There has been a report of successful species classification when used with underwater images (Noman et al., 2021), but no reports with aerial images. To our knowledge, it is a first study using DNN based methods for species level seagrass mapping by remote sensing. The image-to-image translation, used in this study, is one of the applications of DNN. This model is trained with supervised data for transforming the input image into a corresponding output image using the extracted features (Isola et al., 2017). It can be used for semantic segmentation of input images and has also been applied to seagrass bed mapping by remote sensing (Yamakita et al., 2019).

This study aimed to use UAV images and image analysis techniques to create a detailed multispecific seagrass map. The study site was set in a seagrass bed of Saroma-ko Lagoon in northeastern Japan where several seagrass and seaweed species are mixed. We got RGB images by consumer-grade UAV and created a benthic map including the following plant taxa; (1) eelgrass Zostera marina, (2) dwarf eelgrass Z. japonica, (3) green algae (Chaetomorpha crassa, Cladophora sp.), and (4) brown algae (Cystoseira hakodatensis). The accuracies of mapping were compared among three methods, (1) conventional pixel-based supervised classification, (2) object-based supervised classification, and (3) image-to-image translation based on DNN method.

Materials and Methods

In this study, we first undertook UAV photography and transect surveys in the field to create reference data, then conducted image analysis in the laboratory. The overall workflow is shown in Fig. 1.

Figure 1 Methodology workflow of this study.

Parallelograms, rectangles and arrows represent input/output data, data processes and data flows, respectively.

Fieldwork

Fieldwork was carried out on July 9, 2019 at Saroma-ko Lagoon in eastern Hokkaido, Japan (Fig. 2). Saroma-ko Lagoon is a brackish lagoon of about 152 km2 and is connected to the Sea of Okhotsk by two channels, one about 300 m in width and another 50 m. The maximum depth of the lagoon is 19.6 m. Three species of seagrasses (Zostera japonica, Z. marina, and Z. caespitosa) occur along the intertidal and shallow subtidal zones of the lagoon (Biodiversity Center of Japan, 2008). This study was conducted in a seagrass bed at the eastern coast of the lagoon (Fig. 2).

Figure 2 Study site.

(A) Study site is located at Saroma-ko Lagoon in eastern Hokkaido, Japan. (B) Black point indicates the location of this study, black triangles show the channels connecting Saroma-ko Lagoon to the Sea of Okhotsk. (C) Seagrass bed extent along the eastern shore of Saroma-ko. The UAV flight area and cropped area are shown as a red rectangle, transect line is shown as a white solid line. Maps Data: Google, ©2022 Maxar Technologies.

The transect survey and UAV photography were conducted during a low tide. In the transect survey, a transect line was set perpendicular to the shoreline from the shallowest end in the east to the deepest part of the bed in the west until no seagrass appeared (about 600 m offshore). A total of 86 quadrats of 0.25 m2 were placed haphazardly along the transect to cover all present seagrasses and macroalgae along the transect, and species and cover were recorded. Surveys were conducted by wading, snorkeling, and SCUBA diving.

UAV photography was conducted from shore using a quadcopter Mavic2 pro (DJI Co. Ltd., Nanshan, Shenzhen, China). The flight area was set at 580 m offshore and 90 m wide, including a measuring tape used for the transect. We took the images with the RGB sensor camera equipped with the Mavic2 pro at a nadir angle. The flight was automated using DroneDeploy (DroneDeploy Co. Ltd., San Francisco, CA, USA). DroneDeploy enables automatic flight and photography by specifying the flight area, altitude, and overlap rate (front and side) between images. To ensure sufficient spatial resolution for seagrass species identification and to enable orthorectification, we used the setting for DroneDeploy as follows: altitude 30 m, front overlap 80% and side overlap 70%. The camera settings were set before the shooting and were not changed (aperture: f/2.8, shutter speed: 1/400 s, ISO: 200, and white balance: Auto).

Image pre-processing

The captured UAV images were orthorectified using the SfM-MVS processing through the software Metashape ver. 1.7.1 (Agisoft Co. Ltd., Saint Petersburg, Russia). Through SfM-MVS processing, we can produce an orthoimage from overlapped images (Verhoeven et al., 2013). Then, the images were cropped for subsequent analyses. The orthoimage was first converted to a benthic cover map by visual interpretation. Interpreters were the authors who conducted the transect survey and confirmed that vegetation in the study area is uniform along with water depth. Thus, we could distinguish each mapping class by visual inspection without putting quadrats, and applied this method to prepare enough reference data outside of transect survey area for supervised mapping methods. This method was used in a previous study where seagrass areas were recognized based on images (Yamakita et al., 2019).

As a result of the transect survey, three species of seagrass (Z. marina, Z. japonica, Z. caespitosa), green algae (Chaetomorpha crassa, Cladophora sp.), brown algae (Cystoseira hakodatensis), and red algae (Ceramiaceae gen spp.) were observed. Three seagrass species were continuously mixed and the dominant species changed with water depth; Z. japonica (intertidal), Z. marina (shallower subtidal), and Z. caespitosa (deeper subtidal). Zostera caespitosa was difficult to distinguish from Z. marina without observing the belowground part, so the area offshore of 300 m from the shoreline where Z. caespitosa occurred was cropped and excluded from subsequent analysis of orthoimage. This cropping resulted in a total area of 7,884 m2, 291 m along the depth axis and 27 m horizontally to the depth axis. As for macroalgae, red algae were found only in a limited area and were not distinguishable from other vegetation by the naked eye, so they were excluded from the classification. Green algae were combined into one class because it was difficult to distinguish the two species.

These resulted in five benthic classes in this study (Z. marina (ZM), Z. japonica (ZJ), green algae (GA), brown algae (BA), and no vegetation (NV)). RGB values for each class had non-normal, wide-range and overlapped distribution (Fig. 3). Based on these, especially for wide red and green values of ZM and the red value of BA, it was predicted that classification based on color information would be inaccurate. In fact, during visual interpretation, identification by color alone was difficult. However, the patterns of the objects comprising the classes were visible as texture. The interpreters used this information and hand-traced the boundaries of each class on an image editing software, Paint. NET ver.4.2.16 (dotPDN LLC., Seattle, Washington, D.C., USA). The inside of the boundaries was painted and mapped according to class. The orthoimages showing each class are shown in Fig. 4. ZM, ZJ and GA were similarly green, but brightness and visual texture differed. The brightness of ZM ranged widely depending on whether they were submerged or not. We could confirm the stripe pattern of ZM leaves on the orthoimage. Because ZJ leaves were smaller and sparser than ZM, the texture of ZJ was smooth, and color was mixed with the ground under vegetation. The texture of GA was also smooth, but GA was dense and covered with the ground completely. GA was red and string-shaped, thus easily distinguishable. NV was grey and had a white dot pattern of shells.

Figure 3 Frequencies of RGB values for each benthic class.

The histograms show the frequencies of pixel values by color and class. The solid gray vertical line is the average value. Overall, there is a large overlap among classes. The red value in BA are relatively higher.

Figure 4 Orthoimage of each benthic class.

A part of the orthoimage of each class is shown. (A) Z. marina, (B) Z. japonica, (C) green algae, (D) brown algae, and (E) no vegetation. Each benthic class has the unique visual appearance, which enable our visual interpretation.

This study used the maps created by visual interpretation as ground-truth images for training and accuracy verification data. To examine the credibility of the visual interpretation, we compared the ground-truth images with the data obtained from the transect survey. For the comparison, the location of each quadrat was first identified on the orthoimage based on the measurement tape used for the transect installation, and the dominant vegetation classes (ZM, ZJ, GA, BA) were examined. Next, the area corresponding to the quadrat area was cropped from the ground-truth image. The dominant taxonomic classes were examined in the same way and compared with the results of the transect survey. In all cases, however, if the coverage of the dominant class was less than 10%, the no vegetation class (ND) was considered the dominant class.

Mapping method comparison

Mapping by visual interpretation is highly accurate, but requires extensive labor. This study compared three mapping methods (pixel-, object-based classification and image-to-image translation based on DNN) to find a more efficient and reproducible method. All of them are supervised methods, which means that by training the computer using training data, mapping can be done automatically for the rest of the data. In this study, we trained each method using the ground-truth image by visual interpretation. About half of the orthoimage (54%) was used as a training area and the rest (46%) as a validation area, from which accuracy assessment was conducted for each method.

Conventional mapping (Pixel-based and object-based classifications)

Pixel-based and object-based classifications are standard methods for remote sensing images (Dat Pham et al., 2019). These are supervised classifications in which data in some areas are used as training data to classify data in other areas. In this study, the training data were created on ArcGIS pro ver. 2.8.1 (Esri Co. Ltd., Redlands, CA, USA). Pixel-based classification classifies each pixel, while object-based classification classifies each object. An object is a collection of similarly colored neighboring pixels created by the segmentation of the input image. For segmentation, three parameters were adjusted until the object became an appropriate size (Spectral detail: 20, Spatial detail: 5, Minimum segment size: 500 pixels). Spectral detail and spatial detail range from 1 to 20, with higher values used for finer classifications.

The classification parameters were RGB value of pixels for the pixel-based classification, and color and shape of objects for the object-based classification. Parameters describing color and shape of objects are converged color, mean digital number, standard deviation, count of pixels, compactness and rectangularity. These six parameters are optionally available for classification on ArcGIS pro Spatial Analysis Tool.

In this study, the algorithm used for classification was support vector machine (SVM), which was used in seagrass mapping and reported to be sufficiently accurate (Pottier et al., 2021). SVM is not sensitive to training data size and does not assume the probability distribution of the data (Mountrakis, Im & Ogole, 2011). The training data were polygons created from a ground-truth image by uniformly selecting a representative area of each specific class. Pixel-based and object-based classifications were fed different training polygons for higher classification accuracy. The area (number) of training data for each class (ZM, ZJ, GA, BA, and NV) was 75.6 m2 (9), 12.7 m2 (10), 4.04 m2 (5), 0.442 m2 (7) and 21.6 m2 (12) for the pixel-based classification and 75.6 m2 (9), 12.8 m2 (11), 4.31 m2 (7), 0.669 m2 (9) and 15.5 m2 (14) for the object-based classification.

Image translation based on deep learning (pix2pix)

Pix2pix is an image-to-image translation model based on conditional generative adversarial networks (cGANs) (Isola et al., 2017). cGANs are the application of CNN and have two networks: generator and discriminator. The generator transforms the input image, and the discriminator classifies translated image as fake or real by comparing it with the ground-truth image. The generator and discriminator compete with each other, and the generator comes to transform the image into a more realistic one. This model can also be used for remote sensing mapping by translating images to classified images and showed higher accuracy than other deep learning models (Isola et al., 2017). Pix2pix has been applied to various examples, including seagrass mapping for black-and-white aerial photography (Yamakita et al., 2019).

The translation process in pix2pix requires the size of the input image to be 256 × 256 pixels. Therefore, the orthoimage was sliced to an appropriate size without overlap beforehand. Each slice can include one or more classes. After slicing the orthoimage, the numbers of training and validation data were 980 and 840. In general, DNNs are trained more robustly with increasing training data. To prepare a sufficient amount of training data, data augmentation has been carried out in previous studies (Mikolajczyk & Grochowski, 2018; Yamato et al., 2021). Therefore, we added flipped copies to augment the training data. Horizontal, vertical, and simultaneous horizontal and vertical flipped copies of the original training data were added. With this procedure, the number of training data (256 × 256 pixels) increased from 980 to 3920.

GANs-based networks often suffer from a problem called mode collapse (Goodfellow, 2016). This occurs when the training data contain a lot of similar ground-truth images. In such cases, the translated image by the network would also result in similar images. In the study area, the percentage of the ZM area is high, and most of ground-truth data of the training data are dominated by ZM only, which can cause mode collapse. We divided the training data ZM into three subclasses to solve this problem. We reduced colors in the orthoimage of the ZM area to three by posterization and assigned a subclass to each of them. This prevented the homogenization of the ground truth image (Fig. 5).

Figure 5 Example of training data which contain only Zostera marina (ZM).

Pixels of ground-truth area assigned to ZM (left) is re-assigned to three subclasses (right; ZM1, ZM2 and ZM3) by posterization.

Accuracy assessment

Accuracy assessment was performed by comparing the mapping results of each method in the validation area with the ground truth data. Five thousand random points were extracted in the validation area, and a confusion matrix was created for each resulting map. The confusion matrix was used to calculate the overall accuracy (OA) and Kappa coefficient (K) for all classes and the user accuracy (UA) and producer accuracy (PA) for individual classes. OA represents the ratio of the pixel classified correctly. K is a statistic value that expresses the degree of agreement between data, taking into account coincidence (Cohen, 1960). K = 0 means that the degree of agreement is equal to that obtained by chance, and positive values indicate a degree of agreement greater than chance, with the maximum value of 1. In general, the relationship between K and strength of agreement is <0.00: poor, 0.00–0.20: slight, 0.21–0.40: fair, 0.41–0.60: moderate, and 0.61–0.80: substantial (Landis & Koch, 1977). UA is the ratio of each class assigned by the correctly classified mapping, and PA is the ratio of each class assigned by ground truth that is correctly classified.

Results

Image pre-processing

The flight time of the UAV photography was 22 min, and 406 out of 534 taken images were used to orthorectification. The remaining 128 images were taken in deep water where the seagrass was submerged entirely, and they did not show any features such as seagrass leaves, rocks, artificial objects. Due to this, the computer couldn’t detect any matching points across the images which were necessary for creating orthoimage.

The spatial resolution of the created orthoimage was 8.13 mm/pix (Fig. 6A). After cropping orthoimage for analysis, the number of quadrats for the transect survey included in the image was 42. Among them, those dominated by Z. marina, Z. japonica, green algae, brown algae, red algae and no vegetation were 18, 11, 4, 1, 3, and 5, respectively. In the ground-truth image, a part of Z. japonica (4/11) and green algae (1/4) were misclassified as adjacent Z. marina or no vegetation. Similarly, the red algae that did not appear in the ground-truth images were classified as Z. marina or no vegetation (Table 1). Overall accuracy and Kappa of visual interpretation data (Fig. 6B) validated by the field data were 0.786 and 0.687, respectively.

Figure 6 Comparisons of the orthomosaic image (A) and the ground-truth image (B).

The ground-truth image was produced by visual interpretation. A white solid line is a boundary between training area (upper) and validation area (lower) for the different mapping methods. ZM, Zostera marina; ZJ, Zostera japonica; GA, Green Algae; BA, Brown Alga; NV, No Vegetation.

Table 1 Confusion matrix evaluating accuracy of visual interpretation based on field data.

	Field data	UA	
	ZM	ZJ	GA	BA	RA	SD	Total	
Visual interpretation	ZM	18	2	1	0	2	1	24	0.750	
ZJ	0	7	0	0	0	0	7	1.000	
GA	0	0	3	0	0	0	3	1.000	
BA	0	0	0	1	0	0	1	1.000	
RA	0	0	0	0	0	0	0	NA	
SD	0	2	0	0	1	4	7	0.571	
Total	18	11	4	1	3	5	42		
PA	1.000	0.636	0.750	1.000	0.000	0.800			
OA	0.786								
K	0.687								
Note:

UA, User Accuracy; PA, Producer Accuracy; OA, Overall Accuracy; K, Kappa coefficient; ZM, Zostera marina; ZJ, Zostera japonica; GA, Green Algae; BA, Brown Alga; NV, No Vegetation.

Mapping and accuracy assessment

The results of mapping generally agreed among the three different methods (Fig. 7). The study site was overall dominated by Z. marina (ZM). Z. japonica (ZJ) occurred mostly at the eastern part with the large gap (no vegetation, or NV). Gaps are also observed in the western part of the site. Green algae (GA) mostly occurred at the eastern part, whereas brown algae (BA) only in small patches scattered along the whole site. Some misidentifications were observed in these classification methods (Fig. 7). For example, small gaps (no vegetation) in deeper parts were not identified by the pixel-based and object-based methods. In contrast, GA in the shallower parts were overestimated by the pixel-based method. When compared visually, the DNN showed the closest result to the ground-truth. Salt-and-pepper phenomena (speckles noise) was found by the pixel-based classification and made the map difficult to read. Most ZJ in object-based classification were undetected and misclassified into ZM. The DNN also misclassified a giant GA patch at shallow and some NV areas into other classes.

Figure 7 Result of mapping by the three different methods.

Maps were produced from the validation area of orthoimage by pixel-based (A), object-based (B) and DNN (C) methods. The ground-truth data is shown in D. ZM, Zostera marina; ZJ, Zostera japonica; GA, Green Algae; BA, Brown Alga; NV, No Vegetation.

Accuracy assessment showed that the values of OA and K were highest for DNN, followed by the object-based, and the pixel-based methods (Table 2). K value for DNN exceeded 0.6, indicating substantial agreement, whereas that for the pixel-based methods was less than 0.2, showing poor fit. Pixel-based method showed lowest accuracy, because speckles are observed overall in the result map (Fig. 7A).

Table 2 Confusion matrices evaluating accuracy of different mapping methods (A: pixel-based, B: object-based, C: DNN) based on the ground-truth data.

A	
		Ground-truth		UA	
ZM	ZJ	GA	BA	NV	Total	
Map	ZM	1,364	83	43	3	253	1,746	0.781	
ZJ	912	178	43	5	280	1,418	0.126	
GA	550	79	36	2	32	699	0.052	
BA	33	2	1	20	39	95	0.211	
NV	380	100	23	2	537	1,042	0.515	
	Total	3,239	442	146	32	1,141	5,000		
PA	0.421	0.403	0.247	0.625	0.471			
OA	0.427							
K	0.178							
B	
	Ground-truth	UA	
ZM	ZJ	GA	BA	NV	Total	
Map	ZM	3,137	387	124	14	691	4,353	0.721	
ZJ	16	9	3	1	23	52	0.173	
GA	37	5	12	0	2	56	0.214	
BA	3	0	0	11	1	15	0.733	
NV	46	41	7	6	424	524	0.809	
Total	3,239	442	146	32	1,141	5,000		
PA	0.969	0.020	0.082	0.344	0.372			
OA	0.719							
K	0.315							
C	
	Ground-truth	UA	
ZM	ZJ	GA	BA	NV	Total	
Map	ZM	3,160	145	45	9	312	3,671	0.861	
ZJ	11	225	6	0	149	391	0.575	
GA	4	6	17	0	6	33	0.515	
BA	6	0	10	16	3	35	0.457	
NV	58	66	68	7	671	870	0.771	
Total	3,239	442	146	32	1,141	5,000		
PA	0.976	0.509	0.116	0.500	0.588			
OA	0.818							
K	0.618							
Note:

UA, User Accuracy; PA, Producer Accuracy; OA, Overall Accuracy; K, Kappa coefficient; ZM, Zostera marina; ZJ, Zostera japonica; GA, Green Algae; BA, Brown Alga; NV, No Vegetation.

Accuracy by species, shown by the values of UA and PA, also varied greatly (Table 2). ZM, which accounted for the most significant percentage of the study area, showed the highest accuracy for every method. However, PA of the pixel-based classification of ZM (0.421) was much lower than UA (0.781), indicating overestimation. ZJ showed low accuracy in the pixel-based and object-based classifications (0.020–0.403), but higher in DNN (>0.5). ZJ mainly misclassified to ZM and NV in the pixel-based and object-based classification, and these methods could hardly discriminate seagrass species. DNN similarly misclassified ZJ to ZM and NV, but a relatively small extent. GA showed lowest accuracy for almost all methods. The UA was highest for the object-based classification (0.733) for BA, while the PA was higher for the pixel-based classification (0.625). NV showed higher UA than PA for all methods, indicating underestimation mainly due to misclassification to ZM.

Discussion

This study shows that the mapping method based on the combined use of UAV photography and DNN-based image-to-image translation is more accurate than conventional methods, especially on species-by-species identification of seagrass and seaweed species in a multispecific seagrass bed.

Previous studies have attempted to discriminate species of seagrass and macroalgae using satellite and aerial images (e.g., Phinn et al., 2008; Kovacs et al., 2018). Although comparisons should be made with caution due to the difference in sites and methods, the accuracy in our study (OA: 0.818) outperforms those by other studies (OA: 0.23 and 0.28 for Phinn et al. (2008), and 0.64–0.69 for Kovacs et al. (2018)). The grain size of our seagrass bed map (8.13 mm/pix) was much higher than these previous studies (2.4 and 4 m/pix for Phinn et al. (2008), and 2–30 m/pix for Kovacs et al. (2018)), indicating that the spatial resolution was a key factor for successful classification of different plants in multispecific seagrass meadows. In this study, however, spatial extent was small (7,884 m2), covering only 0.035% of the seagrass beds in Saroma-ko Lagoon (22.5 km2 in 2015, Hokkaido Aquaculture Promotion Cooperation, 2015). Linear extrapolation indicates that it would take us more than 1,000 h (i.e. >40 days) to cover the whole seagrass bed by this method, which is not practical considering the labor and seasonal changes in the seagrass bed.

To increase the accuracy of seagrass bed mapping, previous studies have used hyperspectral sensors aboard on satellites and aircraft for species discrimination. This is because seagrasses and seaweed species can be discriminated with different spectral reflectance as well as terrestrial plants (Fyfe, 2003). Although light-weight hyperspectral sensors that can be mounted on UAVs have been developed, they have not been widely used yet because they require complex pre- and post-flight operations for analysis (Adão et al., 2017). Furthermore, it may be difficult even with hyperspectral sensors to discriminate closely related congeneric species of seagrass which have similar characteristics of leaf color. Due to this limitation, it is more effective to develop a new discrimination method using information other than color in the visible band with UAV mounted RGB sensors, which are already used in the field of seagrass research. This study used high-resolution orthoimage for mapping, in which the surface pattern of each class can be identified as well as color. However, mapping based on the conventional classification methods using RGB showed very low overall accuracy, and especially for discriminating ZJ which were misclassified to ZM and NV. This suggests that the pixel and segment colors and shape information used as classification parameters were overlapped among the classes. On the other hand, the mapping based on DNN showed higher accuracy than the conventional methods. This highlights the advantage of the DNN method, in which the computer can extract and use much more information than just color and shape information from the UAV images (Albawi, Mohammed & Al-Zawi, 2017). In fact, ZM and ZJ differ in the presence or absence of a stripe pattern, and the DNN might have identified them based on this information. Therefore, image data with limited spectral information can be analyzed in a more sophisticated way by applying DNN.

In contrast to discrimination of different Zostera species, results of green algae (GA) classification were in low accuracy in all methods. When comparing GA visually in the orthoimage of the training and validation areas, the giant patch of green algae in the validation area has a bright green color that is not seen in the training area. This is due to the difference in the species composition of the GA, which is mainly Cladophora sp. in this brighter clump, and mainly darker Chaetomorpha crassa in the rest. Since these green algae were mixed in the patch, it was not easy to separate them into different classes by visual interpretation. The training data did not sufficiently cover the variability of GA, which may be the reason for the low accuracy. This indicates that even when we use DNNs, there is a limit to their versatility, and performance varies by different seagrass and seaweed species. Higher generalizability will be possible by increasing the variety of training data that sufficiently cover the variability in the study area. BA, which had the smallest area in the training data, also had the lowest UA among the classes in the DNN method. The relatively high UA for the object-based classification suggests that the amount of data is insufficient for training the DNN rather than spectral overlap. Therefore, a possible method to improve accuracy is to provide more training data as for GA.

It has been reported in previous studies that the salt-pepper phenomenon, defined as individual pixels classified differently from their neighbors (Yu et al., 2006), reduces the classification accuracy of pixel-based classification for high-resolution images (Feng, Liu & Gong, 2015). The salt-pepper phenomenon is caused by the internal variability within a classification class that appears as noise in the classification results. In this study, the salt-pepper phenomenon was also observed, and it is one of the factors causing the low accuracy of pixel-based classification. On the other hand, the results of object-based classification showed that segmentation suppressed the salt-pepper phenomenon, making it a more suitable method for high-resolution images. Result of DNN also showed no sand-pepper phenomenon.

In this study, we used a ground-truth image produced by visual interpretation for training data to secure the amount of training data. The machine learning algorithm used in this study, SVM, is known to be more sensitive to the quality of training data than its size (Mountrakis, Im & Ogole, 2011). Therefore, the training data is prepared to represent each class in the training area, and we do not need the entire ground-truth image. On the other hand, the importance of the size of training data is confirmed for DNN by the fact that data augmentation is common to improve algorithms in previous studies (Mikolajczyk & Grochowski, 2018). Therefore, the DNN method is inferior in terms of the time and effort required to prepare the training data. In this study, it took only a few hours to prepare the training data for pixel-based and object-based classification, but it took several days for the DNN method. In terms of computational effort, it took about one night for training DNN using a computer equip high-performance GPU, but several minutes for SVM. In addition, the ground-truth images created by visual interpretation contained some errors when validated by the ground-truth data. Currently, there are examples of underwater photo datasets available for seagrass detection (Reus et al., 2018), but there are no available datasets with labeled aerial seagrass images. Therefore, researchers applying DNNs will need to start by creating a dataset by themselves. However, it is still worth considering the application of DNN because it is expected to achieve highly accurate mapping. Acquiring and creating ground-truth data with quality and quantity are future challenges. The training data proportions are biased toward ZM in this study, thus we would have to be cautious in applying trained network to sites with different vegetation proportions. Investigating the generality of the method is also a future work.

Conclusions

This study reports the result of a case study which applied UAV and machine learning techniques including deep learning at multispecific seagrass bed. UAV enables easier acquisition of high spatial resolution data that was previously difficult to obtain by other remote sensing devices. Pixel-based classification is not suitable for mapping due to the salt-pepper phenomenon. Image-to-image translation based on DNN can discriminate seagrass species and macroalgae, and show higher accuracy than conventional classification methods. Our result indicates that DNN is especially useful when we can obtain high-resolution images with conventional cameras with limited spectral range. Some challenges remain, such as limitation in covering wide areas for the mapping, and in labors for preparing ground truth data. Nevertheless, UAV detailed mapping at coastal area enables scientists further biological research of submerged vegetation based on spatial information.

Supplemental Information

Supplemental Information 1 Raw data for Table 1.

Raw data on the validation of ground-truth data based on in situ census. The file consists of metadata (Sheet 1) and data table (Sheet 2).

Click here for additional data file.

Supplemental Information 2 The raw data for Table 2.

Raw data on the validation of mapping results based on the ground-truth data. The file consists of metadata (Sheet 1), data table for the validation of the pixel-based classification (Sheet 2), of the object-based classification (Sheet 3), and of the DNN classification (Sheet 4)

Click here for additional data file.

We thank Mizuho Namba and Minako Abe Ito for their help with field surveys, and K. Sakaguchi and other members of the Aquaculture Fishery Cooperative of Saroma Lake for their logistical support during our field work in Saroma. We highly appreciate Minako Abe Ito for improving the manuscript.

Additional Information and Declarations

Competing Interests

Author Contributions

Data Availability

The authors declare that they have no competing interests.

Satoru Tahara conceived and designed the experiments, performed the experiments, analyzed the data, prepared figures and/or tables, authored or reviewed drafts of the article, and approved the final draft.

Kenji Sudo performed the experiments, analyzed the data, authored or reviewed drafts of the article, and approved the final draft.

Takehisa Yamakita analyzed the data, authored or reviewed drafts of the article, and approved the final draft.

Masahiro Nakaoka conceived and designed the experiments, authored or reviewed drafts of the article, and approved the final draft.

The following information was supplied regarding data availability:

The raw measurements are available in the Supplemental Files.

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
