# Peer review of "Species level mapping of a seagrass bed using an unmanned aerial vehicle and deep learning technique"

_PeerJ, doi:10.7717/peerj.14017_

## Round 0.1 · original submission · Major Revisions

I have received two reviews from your manuscript. Both reviewers saw great merit in your work and made a few comments. I'm also sending mine in the pdf attached. Specifically, they mentioned that the Methods section need more details and R1 points out that it might be difficult to distinguish seagrass species using RGB in places with different species composition. R2 points out some key aspects in the Methods that should be addressed and also recommends care in the interpretation of the results.

Avoid using acronyms in the paper title. Avoid citing figures and Tables in the Discussion.

Remember to prepare a structured abstract when resubmitting your revised version.

·

Basic reporting

The manuscript follows a proper scientific structure and provides justification of the study. In general, the manuscript is well written and communicate effectively for most of it. However, I felt the methods section can be better described and some comments were added in the manuscript file. Some typos were found in the manuscript, so I recommend to double check the manuscript. The manuscript included tables and figures properly presented and described along the text. Enough context was given. Supplementary files of survey data and validation outputs are provided.

Experimental design

The methods are well described, but some comments are added in the manuscript file. Some statements require clarifications. Descriptions provided in the methods section are enough for allowing replicability. Instruments, software, algorithms, and processes were described. Some methods used are supported by previous studies and cited in the manuscript.

Validity of the findings

This study compares three methods for species specific seagrass mapping using RGB images collected by UAVs. The accuracy of classifiers indicated DNN as an acceptable method for seagrass mapping. Authors discussed limitations of the DNN classifier in larger spatial scales, which would take longer to prepare data, classify, and collect ground-truth data. This might be a good low cost method for mapping seagrass at high resolutions. However, might be difficult to discriminate seagrass species using RGB images from other regions with different seagrass compositions.

Additional comments

Specific comments made to the manuscript are in the attached pdf file.

·

Basic reporting

The paper is, overall, an organized and novel approach to the issue of species-level mapping of seagrass with the use of UAV imagery. The language of the manuscript is clear, though certain expressions can be improved. The literature references are balanced, and appropriately discussed to give the context for the study. The article is well structured, data comprehensible, though clarity of figures could be improved. The results were adequately represented to support their hypothesis, but could be improved as detailed below.

Experimental design

1) The hypothesis of the paper revolves around the ability of advanced image processes and high resolution images to enable the discrimination among different species of vegetation. However, it is important to explain the rationale of the class scheme design from the image processing point of view. It is understood, as the author has discussed in the discussion section, spectral reflectance of the classes in interest could be overly similar, or overly diverse, affecting the classification accuracy. Therefore, description of the classes in terms of spectral responses, visual appearance, texture, perhaps accompanied with images, might later benefit the discussion.
2) Why might conventional approaches fail to detect ZJ class while CNN could?
3) What are the spectral and textural signatures of each class, especially in the Algae? Do spectral responses of each class follow a normal distribution, meaning each class does not contain multiple classes within them?

Validity of the findings

4) As for the description of the object-based classification, it might be clearer to describe the parameters to represent the spatial and textural information of the objects. (contrast, perimeter, etc.) for better reproducibility.

5) The verbal description of the figure 5 could be enhanced to guide the readers to the results presented in the figure.

6) The number of training points and validation for ZM class is disproportionate to other classes, this may result in the training of the classifier being biased towards the ZM class, resulting in misclassification. Furthermore, the Overall Accuracy may be disproportionately skewed toward the accuracy for classifying ZM class, which may undermine the implication of the result. Perhaps the discussion on this point, or consideration of a stratified sampling approach might be beneficial.

Additional comments

6) Minor English expressions may see improvements. "which have too low resolution " , " can be applied to discrimination of seagrass"

---

## Round 0.2 · Minor Revisions

I have now heard back from two of the original reviewers and I'm glad to tell you that they believe the manuscript has greatly improved after revision. I agree with them.

They pointed out a few changes that still need to be done, and I expect it won't take long to do.

I hope to see the revised version soon.

·

Basic reporting

The article follows a proper and professional structure for a scientific journal. The abstract is well segmented and provide a general context of the study. The introduction provides background on context, technologies, and techniques for seagrass mapping, and mention relevant studies. They emphasizes in the use of UAVs and deep neural networks for species specific seagrass mapping at small spatial scale in Japan. The methods are well structured and mention relevant details for reproducing this study. They describe the study site, devices used, techniques for data collection, and software for data processing and analysis. The results section present findings on the UAV images used for creating an orthomosaic, seagrass maps produced by three different techniques, and mapping accuracies. In the discussion section the authors compare their results with other relevant studies. Specifically, they discuss about mapping accuracies, performance of mapping techniques, limitations on the use of RGB images taken by UAVs for discrimination of seagrass species, imagery quality, and quality and quantity of the ground-truth data.
Finally, seven figures and two tables are presented in correct format and referenced in the manuscript. Literature references are presented and cited correctly with style.

Experimental design

The study presented coherent methods to address their research interest. They described the materials and software used, the study site is described, how data was collected is clear, and processing and analyses provided details for replication.

Validity of the findings

The study presented another approach for mapping seagrass species with higher detail. These findings are applicable to small/local regions for seagrass mapping at higher pixel scales with the use of UAV imagery. There are limitations on the use of RGB imagery and ground-truth data, which is important to improve classifier outputs. The authors concluded that the techniques presented here are expected to fill temporal and spatial gaps by satellite imagery at low cost, and allow scientist to use these technologies for further research on submerged vegetation.

Additional comments

Please review these minor comments:

Line 29: replace "monitorings" to "monitoring"
Line 76: replace "WorldView2" to "WorldView-2"
Line 314: were there few matching points for creating the orthomosaic? please clarify here.

·

Basic reporting

Overall, the paper has been much improved in its clarity. If I may add a few minor comments:

1. Other publications using Deep learning methods for classification:
There have been a few publications about deep learning based classification methods for species level seagrass mapping in recent years. Reference to those publications in the introduction and comparison with those research in the discussion is appreciated, as it would enhance the originality of the current paper.

Experimental design

I appreciated the updates and improvements in response to the previous comments, I could understand more clearly the details of your research. To follow up, I would like to add a comment.
2. Description and discussion of figure 3
Since it is important for the methodology and explanation of the confusion of Brown Algae class, the explanation of Figure 3 and its implication could be more elaborated. Does each class have a normal distribution, or is the distribution skewed? Brown Algae class seems to have a bimodal distribution, does it suggest there are more than one spectral class included in this information class? How might the analysis of the spectral response of each class explain the error in classification?

3. It's a minor detail but the white balance information could be included for the UAV parameters.

Validity of the findings

4. Conclusion
The conclusion is a little general, could be more concrete with specific findings from the paper.

---

## Round 0.3 · accepted · Accept

Thank you for making these final adjustments to the manuscript. I believe they fully cover mine and the reviewers' concerns and I’m happy to recommend it for publication